# Review of International Clinical Guidelines Related to Prenatal Screening during Monochorionic Pregnancies

**DOI:** 10.3390/jcm10051128

**Published:** 2021-03-08

**Authors:** Lauren Nicholas, Rebecca Fischbein, Stephanie Ernst-Milner, Roshni Wani

**Affiliations:** 1Department of Social Sciences, D’Youville College, 591 Niagara Street, Buffalo, NY 14201, USA; 2Department of Family and Community Medicine, Northeast Ohio Medical University, 4209 State Route 44, P.O. Box 95, Rootstown, OH 44272, USA; rfischbein@neomed.edu (R.F.); rwani@neomed.edu (R.W.); 3Twin Anemia Polycythemia Sequence (TAPS) Support Foundation, Founder, 1326HS Almere, The Netherlands; stephanie@tapssupport.com

**Keywords:** monochorionic twin pregnancy, screening, clinical guidelines

## Abstract

We conducted a search for international clinical guidelines related to prenatal screening during monochorionic pregnancies. We found 25 resources from 13 countries/regions and extracted information related to general screening as well as screening related to specific monochorionic complications, including twin-twin transfusion syndrome (TTTS), selective fetal growth restriction (SFGR), and twin anemia-polycythemia sequence (TAPS). Findings reveal universal recommendation for the early establishment of chorionicity. Near-universal recommendation was found for bi-weekly ultrasounds beginning around gestational week 16; routine TTTS and SFGR surveillance comprised of regularly assessing fetal growth, amniotic fluids, and bladder visibility; and fetal anatomical scanning between gestational weeks 18–22. Conflicting recommendation was found for nuchal translucency screening; second-trimester scanning for cervical length; routine TAPS screening; and routine umbilical artery, umbilical vein, and ductus venosus assessment. We conclude that across international agencies and organizations, clinical guidelines related to monochorionic prenatal screening vary considerably. This discord raises concerns related to equitable access to evidence-based monochorionic prenatal care; the ability to create reliable international datasets to help improve the quality of monochorionic research; and the promotion of patient safety and best monochorionic outcomes. Patients globally may benefit from the coming together of international bodies to develop inclusive universal monochorionic prenatal screening standards.

## 1. Introduction

Clinical guidelines serve to optimize the care of patients by assisting clinicians and other healthcare professionals [1]. Based on the latest and best available scientific knowledge and, where evidence is scarce, consensus opinion of the experts from the respective field, evidence-based clinical guidelines represent an important step toward the dissemination and implementation of evidence-based treatments into clinical practice [2] and can directly influence the quality of patient care [3]. A recent review examined eight international guidelines related to management of twin pregnancies and found consensus among the guidelines in the areas of (1) first trimester screening including assessment of gestational age as well as identification of chorionicity and amnionicity, (2) nuchal translucency and anatomy screenings, and (3) biweekly ultrasounds for monochorionic and every 4th week for dichorionic pregnancies [4]. Areas of disagreement among the guidelines included utility of cervical length scans for to screen for preterm birth, fetal growth discordance screening, and routine performance of MCA-PSV and UA doppler at every ultrasound scan [4]. However, given the high risk nature of monochorionic twin pregnancies and the possibility of complications [5], further attention is required to understand how national and international guidelines compare with regard to prenatal screening for monochorionic twin pregnancies. Within the topic of monochorionic pregnancy, there exist several internationally dispersed clinical guidelines related to prenatal screening.

Prenatal screening, particularly the use of ultrasonography, is imperative in monochorionic pregnancies, which have long been fraught with a mortality rate that exceeds dichorionic by over seven times [2]. In addition, the incidence of congenital anomalies in monochorionic twin pregnancy is increased by >2-fold over dichorionic twins [6] and 6 to 10-fold over singletons [7]. In monochorionic pregnancies, serial fetal ultrasound examinations are necessary to monitor for development of TTTS and TAPS, as well as SFGR, because these disorders collectively affect 15 to 20 percent of monochorionic gestations, have high morbidity and mortality, and are amenable to interventions that can reduce morbidity and mortality [8]. In monochorionic pregnancy, ultrasound is not only what determines a diagnosis (or diagnoses), but, in the case of TTTS, frequency of ultrasounds is also associated with disease severity at the time of diagnosis. For instance, research shows that women who receive less frequent than bi-weekly ultrasounds are more likely to have advanced stages of TTTS upon diagnosis [9].

In 2010, Doctors Moise and Johnson authored a groundbreaking paper entitled, “There is NO diagnosis of twins” [5]. Within their paper they argue that, from a prenatal screening perspective, monochorionic twins are fundamentally different than dichorionic twins. At that time the authors were urging the American Congress of Obstetrics and Gynecology to establish a prenatal screening standard wherein all monochorionic twins receive bi-weekly ultrasounds beginning in gestational week 16 to provide timely detection of monochorionic compilations and better intervention options [5]. Moise would go on to author another paper in 2014 where he more distinctly specified the host of prenatal screenings (umbilical artery, ductus venosus, MCA-PSV) most advantageous to monochorionic pregnancies [10]. Many of these recommendations can be found in the more recent clinical guidelines discussed here. 

The research surrounding monochorionic pregnancy, its associated disorders, and their treatments frequently updates and changes, therefore influencing prenatal screening recommendations. The purpose of this study was to review current international clinical guidelines related to prenatal screening during monochorionic pregnancies. Specifically, to evaluate where they converge and diverge.

## 2. Methods

### 2.1. Search Strategy

Between June through October 2020, we conducted a search for international clinical guidelines related to prenatal screening during monochorionic pregnancies (see Table 1). We performed searches of databases focused on international guidelines as well as published literature. In order to keep the search broad, we used only general keywords in the searches, including combinations of “twin pregnancy” “or” “monochorionic pregnancy” “or” “multiple pregnancy” with the keyword “guideline.” We also reviewed the websites of agencies responsible for guideline creation as well as professional societies related to the management of monochorionic pregnancies. 

### 2.2. Criteria for Inclusion

To be included in this review, content must provide clinical guidance related to screening or surveillance for prenatal screening during monochorionic pregnancies. Guidelines could be related to screening (1) during monochorionic pregnancies in general, or (2) for specific complications that typically only occur during monochorionic pregnancies (e.g., TTTS, TAPS, SFGR). Given the quickly advancing nature of research and practice in this field, we limited the search to content published within the last 10 years. Additionally, because we are interested in comparing international guidelines, we included guidelines regardless of language and used online translation software to review titles and abstracts/summaries. We also used online translation software to review full-text materials, and this translation was verified with human translators if the guideline was included in this review.

### 2.3. Criteria for Exclusion

Material was excluded from the review if content was: primary research study, a summary of existing guidelines, letter to editor or commentary, systematic review without resulting guidelines, a repeated publication of guideline in alternate journal or language, or prior version of an updated guideline. 

### 2.4. Selection

Two reviewers were responsible for examining the abstracts or summaries of all the references gathered from the database reviews. Summaries published in other languages were translated to English using an online translator. Full-text versions were obtained for sources that passed the first round of review, with an online translator once again used for languages other than English. Two reviewers examined the full-text to ensure fit with inclusion criteria. 

### 2.5. Data Extraction

We extracted data from the guidelines based on (1) information related to the guideline (e.g., country, most recent update, type of source), (2) recommendations related to screening during uncomplicated monochorionic pregnancies (e.g., first-trimester screening, umbilical cord insertions, nuchal translucency), (3) recommendations related to screening for signs of specific complications including SFGR, TTTS, and TAPS. We determined topics for extraction based on a review of the literature as well as review and approval from subject matter experts. One reviewer extracted data from each guideline, and a second reviewer assessed and updated, as needed, the work of the first reviewer. 

## 3. Results

### 3.1. Description of Sources

Our search resulted in a total of 621 titles from all databases and sources, condensed to 554 when duplicates were removed. A total of 78 materials remained after conducting the title review, which reduced to 55 after the abstract review. Finally, after completion of the full-text review, 25 guidelines from 13 different regions/countries were retained (see Table 2 and Table 3) [8,11,12,13,14,15,16,17,18,19,20,21,22,23,24,25,26,27,28,29,30,31,32,33,34]. Two organizations (NAFTNET, UpToDate) published multiple guidelines on different aspects of monochorionic screening, and we presented their information together. The United States had the most guidelines produced by six unique organizations/authors, followed by the United Kingdom and international associations/authors (three each). The median year of guideline publication/update was 2016 ranging from 2011 to 2020. Most publications were the product of organizations or societies (88%); however, three were produced by independent authors. 

The majority of guidelines (88%) were related to general surveillance of monochorionic pregnancies; however, five focused on specific screening aspects or particular complications such as TTTS [8,18,24,30,32]. Nearly half (44%) of the resources provided their recommendations separated by the quality of evidence using GRADE or a similar system [12,13,14,17,20,21,26,27,32,34].

### 3.2. Universal (or Near) Screening Recommendations 

Only the establishment of chorionicity and amnionicity, often with the caveat of identification as early as possible, was universally mentioned in all sources. Nearly all resources recommended biweekly ultrasound scans starting around gestational week 16. This included checking for SFGR and TTTS complications by regularly assessing growth, amniotic fluids, and bladder visibility. Most resources also reported the importance of conducting the fetal anatomical scan around weeks ranging from weeks 18–22, given the higher incidence of congenital anomalies among monochorionic twins. 

### 3.3. Conflicting Recommendations 

While the majority of resources also mentioned nuchal translucency (NT) screenings (Table 2), some noted that NT discordance might be interpreted as an early sign of SFGR and/or TTTS, which could complicate the typical interpretation of NT screening results [12,17,23,32,33]. However, others commented that early NT discordance is not considered predictive of TTTS and should not be treated as such [26,27]. Again, noting the increased frequency of congenital anomalies among monochorionic twins, some guidelines reported that NT discordance may indicate a chromosomal anomaly.

Another identified conflicting recommendation was related to second-trimester screening for cervical length. Some references reported universal screening should be conducted [15,17,20,25,28,34] around week 20 [25]. One guideline stated that evidence is inconclusive regarding this screening but still endorsed performing this scan [13] while another reported that it may be informative in the presence of preexisting risk factors [31]. Others recommended against universal screening [14,19,26,27,29] indicating that there is no evidence of effective intervention to prevent preterm birth in twins. Some guidelines further suggested that this should not be conducted in cases complicated by TTTS [32] or in either asymptomatic or symptomatic women [33].

Approximately half (*n* = 13, 52%) of the resources recommended routine screening for the complication of TAPS, via MCA-PSV Doppler, as part of the biweekly ultrasound regimen [8,11,12,15,16,17,20,23,25,28,29,30,33]. Obtaining MCA-PSV multiples of the normal median (MoM) values of <1.0 and >1.5 [17,20,25,27], indicating risk of TAPS, were generally reported, although some reported looking for the larger MoM range of <0.8 and >1.5 [8,11,20,29,31]. Recommendations for when to start screening for TAPS differed as well, ranging from 16 weeks to 28 weeks [8,12,26,29,30]. Several sources reported that screening for TAPS should only occur post-TTTS laser surgery [31] or if other complications are occurring [26,27] and should not be part of routine screening. Some resources reported that screening was controversial [29] and could not make a statement for or against TAPS screening [22] or the use of calculation of MCA-PSV MoM [23]. However, still other resources indicated that screening for TAPS at any time using MCA-PSV Doppler has not been shown to improve outcomes and therefore could not be recommended [32].

Similarly, explicit recommendations related to routine inclusion of scans of umbilical artery [8,11,12,15,16,17,20,25,27,29,30,34], umbilical vein [8,15,20] and ductus venosus [8,11,15,16,20,21,27] differed across guidelines. Several guidelines mentioned review of these scans only in context of TTTS and SFGR diagnosis and staging [13,17,18,25,26,31,32,33]. Others indicated that these scans should only be performed if complications are suspected [13,23,26,27,28,29,30,32,33].

We noted that the year of recency of the publication or update appears related to recommendations (Table 4). Specifically, more recent publications promoted screening certain topics while older publications either had no mention of, only recommend with complications present, or recommended against, screening.

## 4. Discussion

The main purpose of this study was to review international clinical guidelines related to prenatal screening during monochorionic pregnancies. Clinical guidelines can directly influence the quality of patient care, ref [3] and so we sought to determine where these monochorionic-specific guidelines converge and diverge.

A comprehensive search for international clinical guidelines related to prenatal screening during monochorionic pregnancies was performed. Guidelines were restricted to publication within the last 10 years and could be related to screening (1) during monochorionic pregnancies in general, or (2) for specific complications that typically only occur during monochorionic pregnancies. Guidelines were included regardless of language, and online translation software was used to review titles and abstracts/summaries; human translators were used for guidelines included in the final review. Two independent reviewers analyzed and extracted relevant content from each included guideline.

Our findings were similar to previous research in terms of general areas of guideline agreement and disagreement [4]. Findings reveal a universal recommendation for the early establishment of chorionicity. Near-universal recommendation was found for bi-weekly ultrasounds beginning around gestational week 16; routine TTTS and SFGR surveillance comprised of regularly assessing fetal growth, amniotic fluids, and bladder visibility; and fetal anatomical scanning between gestational weeks 18–22. Conflicting recommendation was found for nuchal translucency (NT) screening; second trimester scanning for cervical length; routine TAPS screening; and routine umbilical artery, umbilical vein and ductus venosus assessment.

Areas of divergence amongst the guidelines are not entirely surprising given that this is a topic with many quickly developing advancements. For instance, TAPS only became a recognized condition in the year 2007 [35], and only in the past 30 years, with the advent of fetoscopic laser ablation surgery and other treatments, has TTTS no longer been associated with an 80–100% mortality rate [36]. While the median year of guideline publication was 2016, some had not been updated since 2011. How recently a clinical guideline had been published or updated became particularly influential when reviewing the conflicting recommendations. A clear trend was observed with more recent clinical guidelines recommending a given prenatal screening. Guidelines need to be quickly updated and become consistent with the latest available research, ref [37] and even the most recently updated publications failed to mention current opportunities for improved screening. For example, recent research reveals other markers for TAPS have been recorded on ultrasound including: starry sky liver for recipients; ref [38] echogenic placenta for donors; ref [39,40] and cardiomegaly in donors [41]. However, despite at least 86% of TAPS cases showing at least one of these markers, ref [41] no guidelines yet mention these TAPS signs.

Aside of more obvious concerns related to patient care and prenatal outcomes, inconsistencies within monochorionic prenatal screening recommendations work to directly limit monochorionic research efforts. That is, when studying something as rare as monochorionicity, and the even-rarer associated disorders (TTTS, TAPS, SFGR), having the ability to use data from outside a given geographic location becomes vital to the compilation of large, reliable datasets. The ability to collaborate internationally to improve our understanding of monochorionic disorders and the efficacy of their potential treatments should be considered an emergent priority. Previous authors have also made note of this valuable opportunity, stating that international multicenter cooperation can improve knowledge and serve as a base for future trials in MC twins with rare conditions [42]. Consistency in clinical guidelines will help influence clinical practice and consistency in clinical practice related to prenatal screening will help produce more reliable data that can be used to better understand monochorionicity, ultimately improving outcomes.

### 4.1. Limitations and Future Research

This review is not without limitations. While we noted approximately half of the resources used GRADE or a similar system to evaluate the quality of evidence associated with their recommendations, we did not compare across the guidelines in terms of their level of evidence for each recommendation. Future research may explore a limited number of topics by level of evidence, particularly those which are newer and/or have mixed evidence. Additionally, most of the resources were limited to Western and English language guidelines and resources. We were unable to locate results for guidelines in several large regions including Africa, most of Asia, Latin America, and the Middle East. Since we used English-powered academic search engines (e.g., PubMed), our inability to find these guidelines likely represents a limitation of the search tools we used as well as our search methods, rather than a lack of guidelines. Future searches should incorporate collaborators across additional regions who can expand this search and explore differences in surveillance for these pregnancies.

The level of inconsistency amongst the clinical guidelines included for review is notable, especially given the fact that most of the countries included for review are high income countries with similar enough characteristics as related to their ability to provide comprehensive, evidence-based prenatal screening. We suspect that if we had the ability to include these low-medium income countries, we would observe even more inconsistency albeit for more varied reasons.

Not all guidelines are created equally, and we recognize that there are limitations created by economic circumstances, access to educational resources, and the geographic/population disbursement of some countries. It is essential to understand that these factors can have a direct impact on prenatal screening recommendations; however, overseeing bodies should take steps to ensure the highest possible standard of care is recommended. Our results strongly suggest that the first step to doing this is simply frequent review and keeping clinical guidelines current with emerging evidence.

In some countries, such as the United States, insurance issues subvert the provision of correct screening protocols. In this case, the role of clinical guidelines to establish a base level of expected prenatal care becomes even more important.

We recognize that guidelines are not an absolute standard of care, but rather the minimum standard of care as established by overseeing bodies using the available evidence and resources. In addition to establishing clinical guidelines, overseeing bodies should also be ensuring that their members provide this minimum.

Further, we did not include in our review surveillance for other complications of monochorionic twin pregnancies such as higher-order multiples, monochorionic-monoamniotic pregnancies, TRAP, or conjoined fetuses. Future research should explore these other, less common complications associated with monochorionic twin pregnancies. Finally, this review only examined surveillance and did not include treatment, particularly for complicated monochorionic pregnancies and future research should explore these topics.

### 4.2. Conclusions

We conclude that across international agencies and organizations, clinical guidelines related to monochorionic prenatal screening vary considerably. In every instance wherein conflicting screening recommendations were observed, the median year of publication was higher for clinical guidelines that included the given prenatal screening and lower for those that did not, highlighting the important role emerging evidence plays in the development of clinical guidelines. The observed inconsistencies raise concerns related to equitable access to evidence-based monochorionic prenatal care; ability to create reliable international datasets to help improve the quality of monochorionic research; and the promotion of patient safety and best monochorionic outcomes. Patients globally may benefit from the coming together of international bodies to develop inclusive universal monochorionic prenatal screening standards.

## Figures and Tables

**Table 1 jcm-10-01128-t001:** Guideline Search Strategy.

Keywords Search	(“Twin Pregnancy” “or” “Monochorionic Pregnancy” “or” “Multiple Pregnancy”) and “Guideline”
International guideline databases searched	AccessMedicine, CURRENT Practice Guidelines in Primary Care 2020 AccessMedicine, CURRENT Practice Guidelines in Inpatient Medicine 2018–2019 ECRI Guidelines Trust GuidelineCentral UpToDate
Literature databases searched	PubMed CINAHL Cochrane Review
Professional society websites searched	SOGC (Canadian), NVOG (Dutch), FIOG (ultrasound group), NAFNET, FGO, RANZCOG, ACR (USA) RCPI (Rep. of Ireland), GBCOG (Germany), ISUOG (International), RCOG, SMFM and ACOG, CNGOF (France), WAPM, UpToDate, Mexican College of Obstetrics and Gynecology, ItaCOG, SLCOG

**Table 2 jcm-10-01128-t002:** Clinical Practice Guidelines for General Screening during Monochorionic Pregnancy.

Guideline Organization or Author and Year	Dating of Pregnancy/Gestational Age	First Trimester Screening	Establish Chorionicity and Amnionicity	Nuchal Translucency Screening	Twin Labeling	Umbilical Cord Insertions Screening	Placenta Placement	Ultrasound Frequency in Weeks	Fetal Size/Fetal Growth	Screening for Chromosomal Abnormalities	Fetal Anatomical Survey	Fetal Echocardiogram	Uterine Artery Doppler (for Preeclampsia)	Cervical Length
Netherlands														
NVOG [11], (2018)		✓	✓ Between 7–14 weeks	✓	✓			✓ Biweekly from 14 weeks	✓	✓				
Australia and New Zealand														
RANZCOG [12], (2017)		✓	✓ Within the first trimester	✓ discordance can be indicative of TTTS or SFGR				✓ Biweekly starting at 16 weeks	✓	✓	✓ “early anatomy” between 11–14 weeks			
Canada														
SOGC [13], (2017)	✓ first trimester; use the larger fetus to avoid missing SFGR	✓	✓ first trimester	✓		mentioned but no recommendation	✓ as part of 1st trimester ultrasound	? “insufficient evidence” but endorses every 2–3 weeks starting at 16 weeks	✓	✓	✓ Notes lack of evidence to repeat anatomical scan after first normal survey			? “inconclusive studies” but endorses cervical length screening
France														
CNGOF [14], (2011)		✓	✓ Between 7–14 weeks	✓				✓ twice monthly and monthly prenatal evaluations	Only mentioned in context of dichorionic twins	? Mixed evidence regarding use of serum markers in first and second trimester				✕ transvaginal ultrasound predictive of preterm delivery but no intervention effective for asymptomatic pregnancy
Germany														
GG, UM [15], (2019)	✓	✓	✓ By 13 + 6 weeks	✓	✓	✓ For identification purposes	✓ For identification purposes	✓ Every 2 weeks starting at 16	✓	✓	✓ 18–22 weeks	✓ 18–22 weeks		✓
International														
FIGO [16], (2019)	✓	✓ For dating and chorionicity	✓ first trimester					✓ Biweekly	✓					
ISUOG [17], (2016)	✓ By 13 + 6 weeks; use larger fetus to avoid missing SFGR	✓	✓ By 13 + 6 weeks	✓ discordance can be indicative of TTTS or SFGR	✓	✓ for identification purposes	✓ As part of labeling	✓ Every 2 weeks starting at week 16	✓	✓	✓ 18–22 weeks	✓		✓
WAPM [18], (2011)	*	*	✓	*	*	Noted as possible factor to influence intertwin blood flow; no recommendation	*	✓ Biweekly	✓	*	*	*	*	Only in context of shorted length may be contraindication for laser surgery
Ireland														
IOGRCPI [19], (2014)		✓ For dating and chorionicity	✓ first trimester					✓ Every 2–3 weeks starting at 16 weeks	✓					✕ not recommended since there is no evidence of intervention to prevent preterm birth in twins
Italy														
S.I.G.O., A.O.G.O.I., AGU [20], (2016)	✓	✓	✓ first trimester	✓	✓	✓	✓ For identification purposes	✓ Biweekly starting at 16 weeks		✓	✓	✓		✓
Mexico														
SS [21], (2013)	✓	✓	✓ first trimester	✓	✓		✓ For identification purposes	✓ First trimester and biweekly from 16–34 weeks	✓ Every 28 days	✓	✓ 18–21 weeks	? “with emphasis on cardiac anatomy” during anatomy scan		
North America														
NAFTNet [22,23] (2015) **		✓ [22,23]	✓ [22,23]	Enlarged NT indicates poor prognosis	✓ [22,23]	For identification Purposes		✓ [22,23] Every 2 weeks starting at 16	✓ [22,23] At least Every 4 weeks		✓ [22,23] 18–22 weeks	✓ [22,23] 18–22 weeks		
Sri Lanka														
S.L.C.O.G. [24], (2013)	✓ use larger fetus to avoid missing SFGR	✓	✓ early in pregnancy; first trimester	*	*	*	*	*	*	*	*	*	*	*
United Kingdom														
Townsend and Khalil [25], (2018)	✓ use larger fetus to avoid missing SFGR	✓ For chorionicity, amnionicity, TRAP, identification	✓ first trimester	✓	✓	noted in context of decision making for TTTS		✓ Every 2 weeks starting at week 16	✓ From week 22	✓	✓ 18–20 + 6 weeks		✓ Every 2 weeks starting at week 16	✓ At 20 weeks
NICE [26], (2019)	✓ first trimester; use larger fetus to avoid missing SFGR	✓	✓ first trimester	✓ notes discordance not early sign of growth restriction	✓			✓ Biweekly starting at 16 weeks (at least 11 appointments)	✓ Every 14 days	✓	✓			✕ no evidence of intervention to prevent preterm birth in twins
RCOG [27], (2016)	✓ first trimester	✓ For viability, gestational age, and exclude malformations	✓ first trimester	✓ Notes should not be used for early sign of TTTS	✓	mentioned but no explicit recommendation		✓ Biweekly starting at 16 weeks through delivery	✓ Biweekly starting at 16 weeks through delivery	✓	✓ Between 18 and 20+6 weeks	! Cardiac views on ultrasound weeks during anatomy scan; echocardiogram if at risk/has TTTS		✕ Routine cervical length screening not advocated at 20–24 weeks scan
United States of America														
Calvo-Garcia [28], (2016)	✓	✓	✓ first trimester		✓	✓	✓	✓ Every 2 weeks starting at 16	✓	Only mention twins tend to have more congenital malformations	✓			✓
UpToDate [8,29,30], (2019–2020) **	✓ [29,30] use the larger fetus if discordant	✓ [8,29,30]	✓ [8,29,30]	✓ [8,29,30]	✓ [29] after 7 weeks	✓ [29,30]	✓ [29]	✓ [8,29,30] biweekly starting at 16–18 week; weekly 32–36 weeks	✓ [8,29,30] Biweekly starting 16–22 weeks then every 4 weeks or 2–4 week [29]	✓ [29] First trimester combined test; ! [30] recommended only in anomalous fetuses	✓ [8,29,30] 16–18 week; early scan 16–18 weeks with detailed scan at 20 weeks	✓ [8,29,30] 18–22 week [29]; 20 week		? [29] “insufficient evidence for routine screening” ✓ [30] 20–22 weeks
Simpson [31], (2015)	✓ by week 14; use larger twin	✓	✓ first trimester	✓		✓ regularly evaluate with color flow imaging	✓	✓ Every 2 weeks beginning in second trimester	✓	✓	✓	✓		! “Universal screening is controversial but helpful for preexisting risk factors”
SMFM [32], (2013)	✓ Notes crown rump discordance can be sign of TTTS	✓ 10–13 weeks	✓ 10–13 weeks	✓ Notes discordance can be sign of TTTS	*	? Mentioned in context of IUGR and sign of TTTS but screening not recommended	*	✓ Biweekly from 16 weeks until delivery should be considered	✓	*	*	✓ in all monochorionic twins	*	✕ screening for cervical length for TTTS cases cannot be recommender; no interventions shown to improve outcomes
S.M.F.M., ACOG [33], (2016)	mentioned but no recommendation	✓ For chorionicity	✓ By late first trimester or early second	✓ note that may be sign of TTTS; complicating screening			✓ described in context of dichorionic twins	✓ every 2 weeks beginning at approximately16 weeks should be considered		✓	✓ described in context of dichorionic twins			✕ not recommended in asymptomatic or symptomatic women; no interventions shown to improve outcomes
ACR [34], (2017)	✓ when crown rump length is between 45–84 mm	✓	✓ early as possible	✓		✓ second trimester	✓ for placenta previa	✓ start 16 weeks; fetal biometry 2–3 weeks;	✓ every 2–3 weeks	✓ 11–14 weeks	✓ 18–22 weeks	✓ all monochorionic second trimester		✓ may be performed second trimester

Notes. Empty cells indicate screening topic not mentioned; ✓ indicates routine screening recommended; **!** indicates screening performed if complication suspected; ✕ indicates screening not recommended; ? indicates mixed evidence. * Guideline is topic or complication specific. Topic may be out of scope of guidelines. ** Multiple guidelines provided by same organization are presented together.

**Table 3 jcm-10-01128-t003:** Clinical Practice Guidelines for Complication-Specific Screening during Monochorionic Pregnancy.

Guideline Organization or Author and Year	Fetal Growth Restriction Screening	Placental Discordance	Umbilical Artery Screening	Umbilical Vein	Ductus Venosus	FGR/IUGR Type	TTTS Screening	MVP Fluids	Bladder Visibility	TTTS Staging	TAPS Screening	MCA-PSV Delta	TAPS Staging
Netherlands													
NVOG [11] (2018)	✓		✓		✓		✓	✓		✓	✓	✓ <0.8 and >1.5	
Australia and New Zealand													
RANZCOG [12], (2017)	✓		✓				✓	✓	✓		✓	✓ After 20 weeks	
Canada													
SOGC [13], (2017)	! increase surveillance only if growth restriction is suspected	mentioned but no recommendation	! “insufficient evidence”; only perform if other complications present		Not explicit; mentioned in context of disease stage		✓	✓ <2 and >8	Not explicit; mentioned in context of diagnosis	✓			
France													
CNGOF [14], (2011)	Mentioned in context of dichorionic twins		mentioned in context of dichorionic twins				✓ twice monthly screening or even weekly	✓ >8 cm or 10 cm and <2 cm	✓				
Germany													
GG, UM [15], (2019)	✓		✓	✓	✓	✓	✓	✓	✓	✓	✓ State bi-weekly but also no consensus on when to start TAPS screening.	✓	✓
International													
FIGO [16], (2019)	✓		✓		✓		✓	✓			✓	✓ Cutoffs not specified	
ISUOG [17], (2016)	✓	mentioned but no recommendation	✓ From 16 weeks	Not explicit, mentioned in context of TAPS staging	Not explicit, mentioned in context of TTTS and TAPS staging	✓	✓	✓ From 16 weeks	✓	✓	✓	✓ From 20 weeks <1.0 and >1.5	✓
WAPM [18], (2011)	✓	*	mentioned in context of TTTS diagnosis; no explicit recommendation	mentioned in context of TTTS diagnosis; no explicit recommendation	mentioned in context of TTTS diagnosis; no explicit recommendation		✓	✓ >8 cm and <2 cm	✓	✓	mentioned as consequence of poor laser surgery; no explicit recommendation	*	*
Ireland													
IOGRCPI [19], (2014)	✓ every 2–3 weeks from week 16						✓ every 2–3 weeks from week 16						
Italy													
S.I.G.O., A.O.G.O.I., AGU [20], (2016)	✓		✓	✓	✓	✓	✓ Biweekly starting at 16 weeks	✓	✓	✓	✓	✓ listed <1.0 and >1.5 under TTTS topic; <0.8 and >1.5 are used under TAPS topic	✓
Mexico													
CNETS [21], (2013)	✓				✓ At time of anatomical screening		✓ Biweekly from 16–24 weeks	✓					
North America													
NAFTNet [22,23], (2015) **	✓ [22,23] Screening as part of check every 2 weeks starting at week 16	✓ [22]	! [23] Only if suspected complication of TTTS, TAPS, IUGR	! [23] Only if suspected complication of TTTS, TAPS, IUGR	! [23] Only if suspected complication of TTTS, TAPS, IUGR	✓ [22] based on doppler ultrasound	✓ [22,23]	✓ [22,23]	✓ [22,23]		? [22] Some NAFTNet centers recommend part of routine surveillance; others feeling insufficient evidence and recommend against✓ [23] as part of check every 2 weeks starting at week 16	? [23] Some NAFTNet centers recommend part of routine surveillance; others feeling insufficient evidence and recommend against	
Sri Lanka													
SLCOG [24], (2013)	*	*	*	*	*	*	*	*	*	*	*	*	*
United Kingdom													
Townsend and Khalil [25], (2018)	✓	✓	✓ starting at week 16 then biweekly		mentioned in context of diagnosis and prediction; no explicit screening recommendation	✓	✓	✓	✓	✓	✓	✓ >1.5 and <1.0	✓
NICE [26], (2019)	✓ Notes should not be done in first trimester		! Only if growth or fluid discordances		only mentioned in context of severe TAPS		✓ Notes should not be done in first trimester	✓	mentioned but no explicit recommendation		! Weekly MCA-PSV from 16 weeks only if other complications are occurring		Not explicitly, describes monitoring for advance stage
RCOG [27], (2016)	✓ every 2 weeks from week 16 until delivery	mentioned but no explicit recommendation	✓		✓	✓	✓ every 2 weeks from week 16 until delivery	✓	✓	✓	! Only screen after laser for TTTS or if otherwise complicated (SFGR)	✓ Implied <1.0 and >1.5	
United States													
Calvo-Garcia [28], (2016)	✓	✓	! only if complications suspected	! only if complications suspected	! only if complications suspected		✓	✓	✓	mentioned but no explicit recommendation	✓	✓	
UpToDate [8,29,30], (2019–2020) **	✓ [8,29,30]	✓ [8,30]	✓ [8,29,30] Starting at 20 weeks, every 2 weeks [30]	✓ [8]	✓ [8] ! [29,30] in context of complications [29]; if Type I SFGR [30]^;^	✓ [29,30]	✓ [8,29,30]	✓ [8,29,30] >10 cm and <2 cm [29]; Starting at 20 weeks, every 2 weeks [30]	✓ [8,29,30] Starting at 20 weeks, every 2 weeks [30]	✓ [8,30]	✓ [8,29,30] starting at 26–28 weeks; MCA-PSV at each ultrasound, every 2–3 weeks	✓ [8,29,30] <0.8 and >1.5 [8,29] Starting at 16 weeks, every 2 weeks [30]; MCA-PSV screening in third trimester is “controversial” [29]	✓ [8]
Simpson [31], (2015)	✓	✓	mentioned in context of SIUGR types			✓		✓ >8 cm and <2 cm		✓	! “Serial surveillance for iatrogenic TAPS should be routine after laser photocoagulation; Screening for spontaneous TAPS in otherwise uncomplicated monochorionic twinsis not routine.”	✓ <0.8 and >1.5	
SMFM [32], (2013)	*	* Mentioned but not explicitly recommended	! “if there is discordance in fluid or growth, is not unreasonable” but unknown utility	* mentioned in context of TTTS stage, no recommendation	✕ “while associated with TTTS and may potentially improve TTTS detection, not recommended as part ofroutine surveillance.”	*	✓	✓ >8 cm and <2 cm	✓	✓	✕ Not recommended at this time; no evidence that monitoring for TAPS improves outcomes	✕ mentioned >1.5 and <1.0; but cannot recommend screening	*
S.M.F.M., ACOG [33], (2016)	! Mentioned in context of ruling out during TTTS		! “No evidence that it is beneficial in the absence of growth or fluid discordance”	Mentioned in context of TTTS stages	Mentioned in context of TTTS stages		✓	✓ >8 cm; <2 cm	Mentioned in context of TTTS stages	✓			
ACR [24], (2017)	✓	✓	✓ biweekly		! Only when discordant growth is present	✓	✓	✓	✓	✓	✓	✓ biweekly	Not specific,” severity can be graded by discordant Dopplers”

Notes. Empty cells indicate screening topic not mentioned; ✓ indicates routine screening recommended; **!** indicates screening performed if complication suspected; ✕ indicates screening not recommended; ? indicates mixed evidence. * Guideline is topic or complication specific. Topic may be out of scope of guidelines. ** Multiple guidelines provided by same organization are presented together.

**Table 4 jcm-10-01128-t004:** Median Year of Publication for Conflicting Guidelines.

	Recommendations for Screening		No Mention/with Complications/Recommendation against Screening	
Screening Recommendation	Median Year	Guidelines	Median Year	Guidelines
Second-trimester scanning for cervical length	2017	[8,13,15,17,20,25,30,31,34]	2015	[11,12,14,16,18,19,21,22,23,24,26,27,29,31,32,33]
Routine TAPS Screening	2018	[8,11,12,15,16,17,20,23,25,28,29,30,34]	2015	[13,14,18,19,21,22,24,26,27,31,32,33]
Routine umbilical artery assessment	2018	[8,11,12,15,17,20,25,27,29,30,34]	2015	[13,14,16,18,19,21,22,23,24,26,28,31,32,33]
Routine umbilical vein assessment	2019	[8,15,20]	2016	[11,12,13,14,16,17,18,19,21,22,23,24,25,26,27,28,29,30,31,32,33,34]
Routine ductus venosus assessment	2019	[8,11,15,16,20,27]	2016	[12,13,14,17,18,19,21,22,23,24,25,26,28,29,30,31,32,33,34]

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
