# Peer review of "Review of International Clinical Guidelines Related to Prenatal Screening during Monochorionic Pregnancies"

_jcm, 2021, doi:10.3390/jcm10051128_

Round 1
Reviewer 1 Report
The topic is very interesting and appropriate. At present, there is an increase in these types of pregnancies. The authors make an adequate approach to the subject. The authors acknowledge in the limitations that guides from countries in Africa, Asia or the Middle East have not been taken into account. The authors indicate that future studies should establish alliances with the scientific societies of these countries for collaboration. However, the authors should address whether, a priori, they believe that there may be large differences between the guidelines of the countries in those parts of the world and the countries that they have consulted. Guides from Latin American countries are also missing. Many of the scientific societies of the countries not consulted can be accessed from the FIGO website. In addition, most of them allow consultation in English.
http://www.fasgo.org.ar/
https://sochog.cl/
http://www.jsog.or.jp
https://sego.es/
http://www.obgyn.org.il
http://www.ssog.org.sa
http://www.fogsi.org
Author Response
Reviewer 1’s Comments:
- “…the authors should address whether, a priori, they believe that there may be large differences between the guidelines of the countries in those parts of the world and the countries that they have consulted.”
Authors’ response:
“We were surprised by the level of inconsistency amongst the clinical guidelines, especially given the fact that most of the countries included for review are high income countries with similar enough characteristics as related to their ability to provide comprehensive, evidence-based prenatal screening. We suspect that if we had the ability to include these low-medium income countries, we would observe even more inconsistency albeit for more varied reasons.
Not all guidelines are created equally, and we recognize that there are limitations created by economic circumstances, access to educational resources, and the geographic/population disbursement of some countries. It is essential to understand that these factors can have a direct impact on prenatal screening recommendations; however, overseeing bodies should take steps to ensure the highest possible standard of care is recommended. Our results strongly suggest that the first step to doing this is simply frequent review and keeping clinical guidelines current with emerging evidence.
In some countries, such as the United States, insurance issues subvert the provision of correct screening protocols. In this case, the role of clinical guidelines to establish a base level of expected prenatal care becomes even more important.
We recognize that guidelines are not an absolute standard of care, but rather the minimum standard of care as established by overseeing bodies using the available evidence and available resources. In addition to establishing clinical guidelines, overseeing bodies should also be ensuring that their members provide this minimum, with emphasis on the fact they can go above and beyond by exercising clinical judgment and keeping abreast of research.
We have added this discussion under section 4.1 in our discussion of limitations.”
- “Guides from Latin American countries are also missing.”
Authors’ response:
“We have added Latin America under section 4.1 in our discussion of limitations.”
- “Many of the scientific societies of the countries not consulted can be accessed from the FIGO website. In addition, most of them allow consultation in English. http://www.fasgo.org.ar/ https://sochog.cl/ http://www.jsog.or.jp https://sego.es/ http://www.obgyn.org.il http://www.ssog.org.sa http://www.fogsi.org”
Authors’ response:
“The above societies were contacted during the revision period, but we have not received responses as yet.
While we were able to obtain many guidelines, we are also aware there are guidelines we missed in our search. A more comprehensive review would show more inconsistencies, highlighting a need for a generalized overview of management as a base guideline.
In lieu of being able to obtain the suggested guidelines, these societies represent countries that are in our discussion of limitations under section 4.1. Please note that Latin America has now been added to that section, as mentioned above.”
Reviewer 2 Report
This is a comprehensive review and comparison of international clinical guidelines related to prenatal screening during monochorionic pregnancies published in the past decade. The search aims to compare and report the similarities and differences between the different guidelines identified, stating that unified international surveillance may contribute to enahnce research in the field and that observed inconsistencies raise concerns related to equitable access to evidence-based monochorionic prenatal care.
The comparison itself a detailed and in depth thorough investigation, and it is apparent that much effort has been put into this publication.
Different guidelines around the world may stem from objective differences in needs and resources.
High income countries allow different accsess to care than low-medium income countries. Even some high income countries with a large area (i.e. Canada) may not be able to cater the very advanced ultrasound studies to every pregnant woman ( for instance advanced Doppler flow studies). These objective hardships that different places in the world face, make it neccesary to accomodate the local guidelines to the feasible care that exists in that area even if less than ideal. Medical insurance coverage also differs in different countries and affect the level of care that is accessible for a parturient. While ideally every pregnant woman around the world should have access to the best available care, in practice they should get the best realistic care.
International societies guidelines may set the highest standards, and local guidelines in turn serve to address the local capability to deliver care in all the above mentioned aspects, as well as to serve as the medicolegal document the local physician is committed to, and therefore are expected to display diversitied among different areas around the world.
Moreover, as mentioned in the text, some of the differences in the guidelines stem from the publication year, for example the need to screen for cervical shortening changed dramatically when an effetive intervention was accepted around 2016, and that is reflected in later guideline publications, As well as recognizing the importance of spontaneous TAPS.
These mount to emphasize the importance of keeping guidelines up to date with literature.
To conclude, local guidelines should be follow the major international societies with adjustments to local needs and ability to deliver care, and should show appropriate divergence. Guidelines are best kept up-to-date and if not, become futile.
Reviewer 3 Report
A well-organised and prepared manuscript about an important problem. Due to its value it should be published in the present form.
The recommendations about prenatal ultrasound of monochorionic twins should be clear and unified to provide the optimum care for these high-risk pregnancies.
Author Response
Reviewer 3’s Comments:
- A well-organised and prepared manuscript about an important problem. Due to its value it should be published in the present form. The recommendations about prenatal ultrasound of monochorionic twins should be clear and unified to provide the optimum care for these high-risk pregnancies.
Authors’ response:
“We thank Reviewer 3 for their comments.”
Round 2
Reviewer 2 Report
The amendments made address the main issued raised adequately/
Next step may be to suggest a format of unified guidelines to help serve the important cause of world wide research collaborations, and adequate care to all patients across different countries.